# Rapid and Non-Destructive Methodology for Measuring Canopy Coverage at an Early Stage and Its Correlation with Physiological and Morphological Traits and Yield in Sugarcane

**Raja Arun Kumar [1,*], Srinivasavedantham Vasantha [1], Raju Gomathi [1], Govindakurup Hemaprabha [2], Srinivasan Alarmelu [2], Venkatarayappa Srinivasa [2], Krishnapriya Vengavasi [1] , Muthalagu Alagupalamuthirsolai [1], Kuppusamy Hari [1], Chinappagounder Palaniswami [1], Krishnasamy Mohanraj [2], Chinnaswamy Appunu [2], Ponnaiyan Geetha [1], Arjun Shaligram Tayade [1,3], Shareef Anusha [1], Vazhakkannadi Vinu [2], Ramanathan Valarmathi [2], Pooja Dhansu [4] and Mintu Ram Meena [4]**

[1] Division of Crop Production, ICAR-Sugarcane Breeding Institute, Coimbatore 641007, India; vasanthavedantham@yahoo.com (S.V.); r.gomathi@icar.gov.in (R.G.); k.vengavasi@icar.gov.in (K.V.); alagu.m@icar.gov.in (M.A.); k.hari@icar.gov.in (K.H.); palaniswami.c@icar.gov.in (C.P.); geetha.p@icar.gov.in (P.G.); arjun.tayade@icar.gov.in (A.S.T.); anusha.s@icar.gov.in (S.A.)

[2] Division of Crop Improvement, ICAR-Sugarcane Breeding Institute, Coimbatore 641007, India; g.hemaprabha@icar.gov.in (G.H.); s.alarmelu@icar.gov.in (S.A.); v.sreenivasa@icar.gov.in (V.S.); k.mohanraj@icar.gov.in (K.M.); c.appunu@icar.gov.in (C.A.); vinu.v@icar.gov.in (V.V.); valarmathi.r@icar.gov.in (R.V.)

[3] ICAR-National Institute of Abiotic Stress Management, Baramati 413115, India

[4] ICAR-Sugarcane Breeding Institute Regional Station, Karnal 132001, India; pooja@icar.gov.in (P.D.); mr.meena@icar.gov.in (M.R.M.)

*   Correspondence: r.arun@icar.gov.in or arunkr.plphy@gmail.com

**Abstract:** Screening for elite sugarcane genotypes for canopy cover in a rapid and non-destructive way is important to accelerate varietal/clonal selection, and little information is available regarding canopy cover and leaf production, leaf area, biomass production, and cane yield in sugarcane crop. In the present investigation, the digital images of sugarcane crop by using *Canopeo* software was assessed for their correlation with the physiological and morphological parameters and cane yield production. The results revealed that among the studied parameters, canopy coverage has shown a significantly better correlation with the plant height (0.581 **), leaf length (0.853 **), leaf width (0.587 **), and leaf area (0.770 **) in commercial sugarcane clones. Two-way cluster analysis has led to the identification of Co 0238, Co 86249, Co 10026, Co 99004, Co 94008, and Co 95020 with better physiological traits for higher sugarcane yield under changing climate. Additionally, in another field experiment with pre-breeding, germplasm, and interspecific hybrid sugarcane clones, the canopy coverage showed a significantly better correlation with germination, shoot count, leaf weight, leaf area index, and plant height, and finally with biomass ($r$ = 0.612 **) and cane yield ($r$ = 0.458 **). It has been found that the plant height, total dry matter (TDM), and leaf area index (LAI) had significant correlation with the cane yield, and the canopy cover data from digital images act as a surrogate for these traits, and further it has been observed that CC had better correlation with cane yield compared to the other physiological traits viz., SPAD, total chlorophyll (TC), and canopy temperature (CT) under ambient conditions. Light interception determined using a line quantum sensor had a significant positive correlation ($r$ = 0.764 **) with canopy coverage, signifying the importance of determining the latter in a non-destructive way in a rapid manner and low cost.

**Keywords:** sugarcane clones; canopy cover; light interception; biomass; cane yield

## 1. Introduction

Sugarcane is one the most important industrial crops in global agriculture, and it has emerged as a multiproduct crop benefiting producers and consumers [1]. Sugarcane is the

second most important industrial crop after cotton in India, occupying about 5 million ha of land with a sugar production of 32.38 metric tons [2]. The sugar industry is the second largest agro-industry in India, and it contributes to 1.1% of the national GDP besides providing for 4% of the population residing in rural areas [3]. Due to the burgeoning population and other constraints (abiotic stress), Ref. [3] the cultivated area of sugarcane will mostly remain static; hence, the only option for the increasing production is to go the vertical way/enhance crop productivity. Sugarcane is a $C_4$ crop that produces four carbon compounds as the primary product in the carbon assimilation cycle, and it is commonly grown from latitude 36.7° N to 31° S and from sea level to 1000 m of altitude, and generally sugarcane grows slowly during the early part of its growing period compared to other tropical gramineous crops, taking up to 4 months to produce a complete leaf canopy which intercepts nearly all the incoming radiation [4–7], while maize (*Zea mays*) and pearl millet (*Pennisetum glaucum*) normally produce a complete leaf canopy within 2 months of sowing [8–10]. Owing to the slow production of a complete leaf canopy, dry biomass production is slow in sugarcane during the early part of the growth period. A comparison of sugarcane and maize made in Zimbabwe [11] showed that the growth in dry mass was faster at 4 months after sowing. Sorghum (*Sorghum bicolor*) and maize grow at similar rates [12], suggesting that sorghum grows faster than sugarcane. On the other hand, Bull and Glasziou [4] showed that early growth in dry mass in sugarcane is slow in most of the regions, and high yields produced by sugarcane are mainly due to an extended growth period rather than superior photosynthetic efficiency.

Canopy cover is a useful trait for monitoring crop productivity [13], and canopy photosynthesis is greatest when the crop reaches its maximum canopy cover to intercept nearly most of the incident light and absorb the required photosynthetic radiation for photo-biochemical processes and yield formation [14,15]. The most common method for measuring canopy cover is by determining the light interception with a line quantum sensor [16,17]. Shepherd et al. [13] reviewed the notion that this system would be time-consuming and costly, as the measurements should be collected near solar noon [16,18].

Another method involves using drone-based digital image capturing and processing to predict the canopy coverage. However, such a facility may not be equally accessible to all in the scientific community.

In this context, a recently developed method of Oklahoma State University for measuring canopy coverage, called *Canopeo*, which rapidly determines the canopy coverage (%) using digital images, employs an application for iOS (Apple) and Android (Google) devices and Matlab (Mathworks) [19]. *Canopeo* (Oklahoma State University App Center, Stillwater, OK, USA) is an automatic colour threshold (ACT) image analysis tool that analyses pixels based on the red-to-green (R/G) and blue-to-green (B/G) colour ratios and an excess green index [13]. *Canopeo* was accurate and faster at computing canopy cover than other software and is widely being used in many crops such as alfalfa, cover crops, soybean, sorghum, wheat, potatoes, and turf grass (https://canopeoapp.com/, accessed on 10 July 2022).

Canopy cover (CC), leaf area, and biomass production are reported to be the most important physiological components resulting in better cane yield in sugarcane; hence, quantification of canopy cover, which is the primary factor for biomass production, is highly essential, and the later requires a lot of labour, resources, and time through a leaf area measurement by destructive sampling or by light interception method using line quantum sensors or by the drone-based image capturing. Several reports are available on various crops regarding canopy cover by *Canopeo*, and little information is available regarding canopy cover, leaf production, leaf area, and biomass production in sugarcane crop. The robustness of the *Canopeo* tool needs to be validated and compared with data generated from line quantum sensor and leaf area measurements for light interception measurements in sugarcane crop. The ICAR-Sugarcane Breeding Institute, Coimbatore, India, a century-old historical institute known for the "Nobilization of cane", has evolved more than 3500 sugarcane clones, and to sustain sugarcane production the canopy coverage (CC) trait is highly essential for screening climate-resilient sugarcane clones. Therefore, the

present investigation was carried out to (i) evaluate sugarcane canopy cover measured with *Canopeo* and with the light interception method using a line quantum sensor to find an association between two different methods and (ii) to analyse the canopy cover in sugarcane including commercial, interspecific hybrids and germplasm clones and to establish its correlation with physiological and morphological parameters, biomass, and cane yield traits in field conditions.

## 2. Materials and Methods

### 2.1. Plant Material and Crop Management of Commercial Hybrids of Sugarcane Clones

Sugarcane clones of commercial hybrids types viz., CoM 0265, Co 86249, Co 99004, Co 10026, Co 86010, CoC 671, Co 1148, Co 95020, Co 2001-13, Co 86032, Co 7717, Co740, Co 62175, Co 8371, Co 0218, CoLK 8102, BO 91, Co 775, Co 0212, Co 91010, ISH 100, Co 94008, Co 0238, Co 86011, Co 8338, Co 85019, Co 8208, Co 419, CoV 92102, Co 13006, and Co 8021 (Table 1) were grown at the ICAR-Sugarcane Breeding Institute, Coimbatore (11°0′34″ N, 76°55′2″ E, 430 m above mean sea level), Tamil Nadu, India. Two budded sets, thirty-eight per row of 6.0 m, were planted, and a full dose of phosphorous ($P_2O_5$) was applied in the furrows before planting as basal fertilization, while nitrogen (N) and potassium ($K_2O$) were applied in two equal measures at 45 days after planting (DAP) and at full earthing-up (90 DAP). Detrashing of dried leaves was done at 5, 7, and 10 months after planting for proper sunlight penetration. The crop stand was free from significant disease or insect damage. The morpho-physiological data, viz., germination percent, leaf length, leaf width, leaf number, leaf area, shoot thickness, and plant height, were determined by following standard procedure.

**Table 1.** Sugarcane clones (source: Hemaprabha et al., 2018) [20].

| No. | Clone [a] | Maturity | Colour | Sucrose (%) |
|-----|-----------|----------|--------|-------------|
| 1 | BO 91 | Mid late | Yellow purple | 16.40 |
| 2 | Co 10026 * | Early | Pinkish yellow orange | 19.42 |
| 3 | Co 13006 | Mid late | Yellow orange | 19.15 |
| 4 | Co 0212 * | Mid late | Purple | 19.67 |
| 5 | Co 0218 | Mid late | Yellow purple | 20.12 |
| 6 | Co 0238 * | Early | Golden purple | 19.25 |
| 7 | Co 1148 | Mid late | Light purple | 15.18 |
| 8 | Co 2001-13 * | Mid late | Purple | 19.03 |
| 9 | Co 419 | Mid late | Dark purple | 17.09 |
| 10 | Co 62175 * | Mid late | Greenish purple | 17.35 |
| 11 | Co 740 | Mid late | Yellowish green | 17.96 |
| 12 | Co 7717 | Early | Purple | 17.90 |
| 13 | Co 775 | Early | Light purple | 18.32 |
| 14 | Co 8021 | Mid late | Purple | 17.86 |
| 15 | Co 8208 | Early | Purplish pink | 17.86 |
| 16 | Co 8338 | Early | Dark purple | 18.82 |
| 17 | Co 8371 | Mid late | Green yellow | 18.18 |
| 18 | Co 85019 * | Mid late | Purple | 16.39 |
| 19 | Co 86010 * | Mid late | Yellowish green | 18.45 |
| 20 | Co 86011 | Early | Purple | 19.98 |
| 21 | Co 86032 * | Mid late | Reddish pink | 19.45 |
| 22 | Co 86249 * | Mid late | Green yellow with purple tinge | 18.82 |
| 23 | Co 91010 * | Mid late | Yellow green with purple tinge | 19.89 |
| 24 | Co 94008 | Early | Purple | 18.71 |

**Table 1.** *Cont.*

| No. | Clone [a] | Maturity | Colour | Sucrose (%) |
|---|---|---|---|---|
| 25 | Co 95020 | Mid late | Yellowish green | 18.79 |
| 26 | Co 99004 | Mid late | Yellowish green | 20.00 |
| 27 | CoC 671 | Early | Light purple to purple yellow | 21.00 |
| 28 | CoLk 8102 | Mid late | Yellowish purple | 18.00 |
| 29 | CoM 0265 | Mid late | Green | 19.33 |
| 30 | CoV 92102 * | Mid late | Purple | 19.80 |
| 31 | ISH 100 | Mid late | Light purple green | 18.20 |

[a] Asterisks (*) indicate clones suitable for drought conditions [1].

### 2.1.1. Germination%, Plant Height, and Shoot Thickness

The number of germinants/row was recorded at 30 DAP, and germination % was derived. The plant height was measured from the base to the top most visible transverse mark on the 60 DAP using a measuring tape and the shoot thickness with a digital vernier calliper (Mitutoyo, Kawasaki, Japan) [21].

### 2.1.2. Leaf Traits

Leaf area (*LA*) was determined in a non-destructive manner by linear measurement method as mentioned by Montgomery (1911):

$$LA = LBK \left( cm^2 \right) \qquad (1)$$

where *L* = maximum length of length, *B* = maximum breadth, and *K* = constant (0.75 based on regression analysis).

### 2.1.3. Biomass

During the formative stage, the biomass samples were collected in a one-meter square area, and all the samples were oven-dried ($60 \pm 5\,°C$) until a constant weight was reached.

### 2.1.4. Determination of Canopy Cover in Commercial Hybrids of Sugarcane Crop at Early Formative Phase

The non-destructive method of canopy coverage was recorded at 60 DAP using *Canopeo* software installed in Android mobile phone. *Canopeo* is an application for iOS (Apple, Cupertino, CA, USA) and Android (Google, Mountain View, CA, USA) mobile devices and Matlab (Mathworks, Natick, MA, USA) that can rapidly analyse canopy cover (Figure 1a) from pictures [19]. The accuracy of the CC recorded through *Canopeo* software is 91% (correctly classify pixels as green/true positive), and specificity is 89% (non-green/true negative) as mentioned by the original author of the software from Oklahoma State University [19]. The distance between the mobile and sugarcane plant while recording the measurements is 80 cm. In order to facilitate easy CC recording, a 35-inch length selfie stick was also used for a few taller clones.

The captured image was processed rapidly through the *Canopeo* software immediately after image acquisition on an Android mobile device, and the derived canopy coverage (%) was saved as a separate folder for further analysis. The canopy coverage image was captured by keeping the mobile parallel to the soil [19]. However, in our study, another method (keeping mobile perpendicular to the soil) was also followed along with the standard method (keeping mobile parallel to the soil), and finally, both methods were compared by correlation to identify the best method/position for capturing the image in the sugarcane crop.

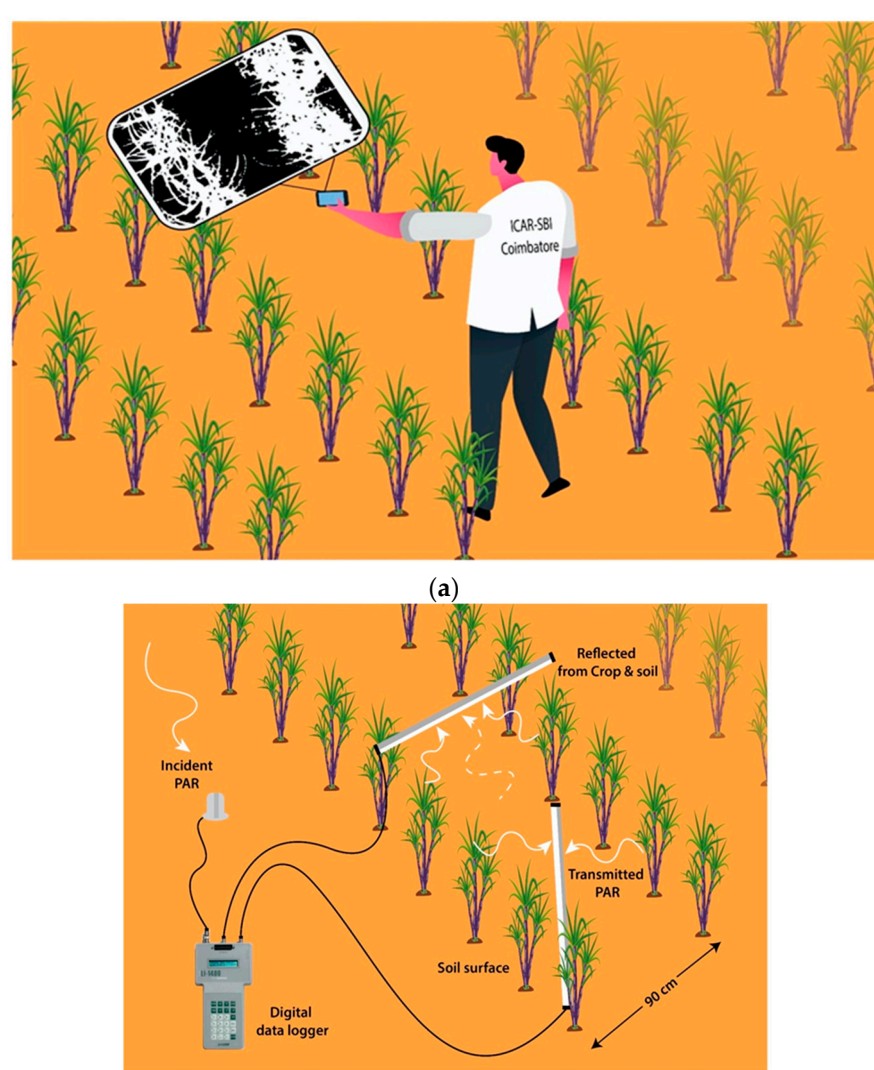

**Figure 1.** (**a**) Representative figure of measuring canopy coverage in a sugarcane field. (**b**) Representative figure of measuring light interception in a sugarcane field.

The samples are recorded for the canopy coverage from 9.00 to 11.00 AM, and data are recorded just opposite to the sunlight direction in order to avoid the shade of the observer. The represented values are the average of four observations per replication, i.e., a total of 8 observations per treatment.

### 2.1.5. Determination of Light Interception in Commercial Hybrids of Sugarcane Crop at Early Formative Phase

The light interception (LI%) was determined using line quantum sensor LI-191SA (LICOR Inc., Lincoln, NE, USA) connected with the LI-1400 a multipurpose datalogger that functions both as a data logging device and a multichannel, auto-ranging meter between 11.00 to 12.00 IST (Figure 1b). The intercepted photosynthetically active radiation (IPAR) for a particular day was computed as the difference between incident PAR at the top and the transmitted PAR received at the bottom of the canopy (the radiation reflected from the crop and soil was also taken into account for deriving the LI%), and the correlation between the CC% and LI % was conducted. Also, the radiation reflected by the soil surface was also determined and finally incorporated for the LI% calculation.

*2.2. Plant Material and Crop Management of the Breeding Population, Interspecific Hybrids, and Basic Species Clones of Sugarcane Clones*

In order to determine the correlation between the canopy coverage with biomass and cane yield, 38 sugarcane clones including improved breeding population clones (004-73, 04-423, 14-154, 07-520, 04-595, 04-472, 12-127, 97-77, 01-807, 20-158, 20-614, 20-335, 99-45, 98-290, WL 10-40, 14-161, 81 GUK 192, 81 GUK 527, 92 GUK 220, 97 GUK 111, 98 GUK 116, GUK 00-910, GUK 02-91, GUK 06-402, 88 GUK 072, 97 GUK 9, 97 GUK 74, 987 GUK 124, 20-191, 07-776, 99-19, 99-291, 06-013, and 01-803), interspecific hybrids (ISH 107 and ISH 111), and germplasm clones (Kheli and Pathri) were planted in the randomized block design in two replications at the experimental farm of ICAR-Sugarcane Breeding Institute, Coimbatore, India (11°0'34'' N, 76°55'2'' E, 430 m above mean sea level) during the years 2021–2022. The canopy coverage was recorded at 60 DAP using *Canopeo* software and analysed as mentioned in Experiment 1. The germination, shoot thickness, and plant height were determined as mentioned in Experiment 1.

During the formative stage, the fresh biomass samples were collected in a one-meter square area, and all the samples were separated into leaf, sheath, and stem parts and were oven-dried (60 ± 5 °C) for determination of constant weight. The constant dry weight was used for computing the overall dry matter production.

### 2.2.1. SPAD

Non-destructive chlorophyll estimation was recorded using a SPAD meter (Soil Plant Analysis Development) (atLeaf, Wilmington, Delaware, NC, USA) that computes the chlorophyll content of a leaf by recording the transmission of red light and infrared light at 660 nm and 940 nm, respectively, and converts the reading into a digital signal [1].

### 2.2.2. Canopy Temperature

The canopy temperature was measured with a thermal imaging infrared camera (FLIR E6) between 11:30 a.m. and 12:00 noon on cloudless days. The image captured was processed through FLIR software (FLIR Tools version 5.1.15036.1001), and the final data were used. The thermal imaging camera was held to view the crop at a 30° angle from horizontal at a 90° angle to the row, with the minimum exposure to the soil, and the emissivity factor of 0.95 was used for the green canopy. Each canopy temperature measurement was the average of three readings at different locations in each clone. Images were registered in the Thermal MSX® mode (FLIR Systems, Wilsonville, OR, USA), and files were saved in standard 14-bit JPG format.

### 2.2.3. Chlorophyll Fluorescence

Chlorophyll fluorescence ($F_v/F_m$) was measured in intact sugarcane leaves using a chlorophyll fluorometer (model OS30p, Opti-Sciences, Hudson, NH, USA). The leaves were dark-adapted for 15 min using leaf clips (Opti Sciences), and the ($F_v/F_m$) readings were recorded by passing a saturating light:

$$\frac{F_v}{F_m} = \frac{F_m - F_o}{F_m} \tag{2}$$

where $F_v/F_m$ = ratio of variable fluorescence to maximal fluorescence, $F_m$ = maximal fluorescence, $F_o$ = minimal fluorescence, and $F_v$ = variable fluorescence of photosystem II [1].

2.2.4. Sucrose, Cane Yield, and CCS

Sugarcane juice was extracted in a crusher with 65% extraction capacity, and the juice quality was analysed as total soluble sugars (TSS) (Brix) and sucrose content (Pol%) according to the standard method [22]. Cane yield was estimated at the 12th month of the crop stage, and the middle 4 rows of canes were harvested and weighed for the plot yield, and the yield per hectare was calculated and expressed as t ha$^{-1}$. Commercial cane sugar (CCS) was determined and expressed in percentage and t ha$^{-1}$ according to Equations (3) and (4), respectively [21].

$$CCS\% = \frac{\text{Sucrose content} \times 1.022}{\text{TSS} \times 0.292} \tag{3}$$

$$CCS\ (t/ha) = \frac{CCS\% \times \text{Yield}}{100} \tag{4}$$

*2.3. Statistical Analysis*

Analysis of variance (ANOVA) was performed on the data following the method of Gomez and Gomez (1984) [23], and the least significant difference (LSD) values were calculated at the 5% probability level. Duncan multiple range test (DMRT) was performed to separate significant genotypes, and alphabets were superscripted for easy view. The Pearson-product-moment correlation coefficients (*r*) between leaf length, leaf width, leaf number, leaf area, shoot thickness, plant height, germination percent, and canopy cover were computed using SAS 9.3 (SAS Institute, Cary, NC, USA) [24]. The scatterplot matrix showing the correlation and frequency counts among the studied parameter, i.e., canopy coverage % (CC), cane yield (CY), dry biomass at formative phase (DWFP), shoot diameter (SD), leaf area (3rd top visible dewlap), leaf area (cm$^2$), leaf width (cm), leaf length (cm), leaf number (L.No), and plant height (PH), was created using JMP genomics software version 6.1. Two-way cluster analysis with the wards method showing the grouping of sugarcane clones with the studied parameter was also conducted using JMP genomics software. Regression analysis was carried out between the CC and biomass, cane yield, and their corresponding slope (*β*), and significance was determined following "F" test at 0.05 probability. A correlation diagram displaying the correlation between the studied parameters along with the *p*-value was conducted through R software version 4.1.3.

### 3. Results

*3.1. Association between Sugarcane Canopy Cover Measured with Canopeo at Different Positions*

The canopy cover (CC) data was determined at the early formative phase (60–150 DAP) through *Canopeo*, and both digital images acquired parallel to the ground and perpendicular to the ground were analysed for their association and relevance in sugarcane crop. Among the four phases of the sugarcane crop, the formative phase which starts at 60 DAP is reported to have high relevance to the cane yield; hence, the CC data were recorded at 60 DAP. The correlation between canopy coverage (%) from an image acquired parallel to the ground and perpendicular to the ground is shown in Figure 2a. A significantly better correlation of r = 0.870 ** was observed between the canopy coverage (%) data through images acquired parallel to the ground and perpendicular to the ground of the sugarcane crop. Canopy cover images were taken in properly weeded/weed-free fields to reduce the data error; i.e., the background images of weeds mimic the crop, and this results in an overestimation of CC data. The data revealed a significant linear relationship at 1% probability level between the data captured through two positions.

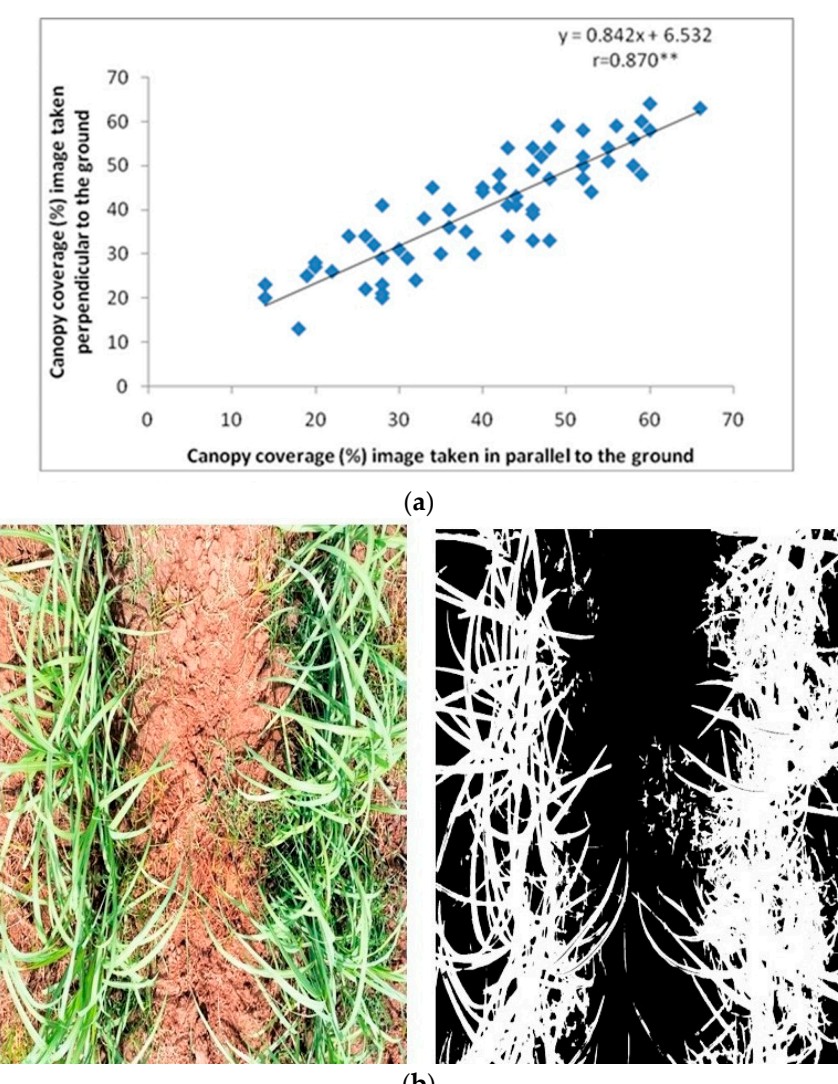

(**a**)

(**b**)

**Figure 2.** (**a**) Correlation between canopy coverage (%) image acquired parallel to the ground and perpendicular to the ground. ** denotes significant at 1%. (**b**) Canopy coverage (%) images, i.e., original image (left) and classified image (right) of sugarcane clone.

*3.2. Association between Sugarcane Canopy Cover Measured with Canopeo and with the Light Interception by PAR (Photosynthetically Active Radiation) Line Quantum Sensor*

The light interception (LI) data were recorded simultaneously while capturing the canopy cover images through *Canopeo* using Android mobile. Light interception data were acquired through multi-channelled PAR quantum sensors; i.e., one line quantum sensor was placed diagonally between the rows of sugarcane crop, and another line quantum sensor between and above the crop for measuring the transmitted PAR and reflected PAR simultaneously. Incident PAR measurement was achieved through a point sensor for ease of work.

Further, a significant correlation (r = 0.764 **) was observed between canopy coverage (%) from images acquired in parallel to the ground, and light interception by a line quantum sensor (Figure 3) confirms the accuracy of the CC data of *Canopeo*. A positive coefficient indicates that as the value of the independent variable (canopy cover) increases, the mean of the dependent variable (light interception) also tends to increase. The slope coefficient or *β* value of the regression was 0.695, and the coefficient represents the mean increase of LI% for every additional increment of CC. In the present regression equation, (Figure 3) for every 0.695 increment in CC, a correspondingly one unit increment in LI was observed, and the model was found statistically significant at 1% probability through the "F" test.

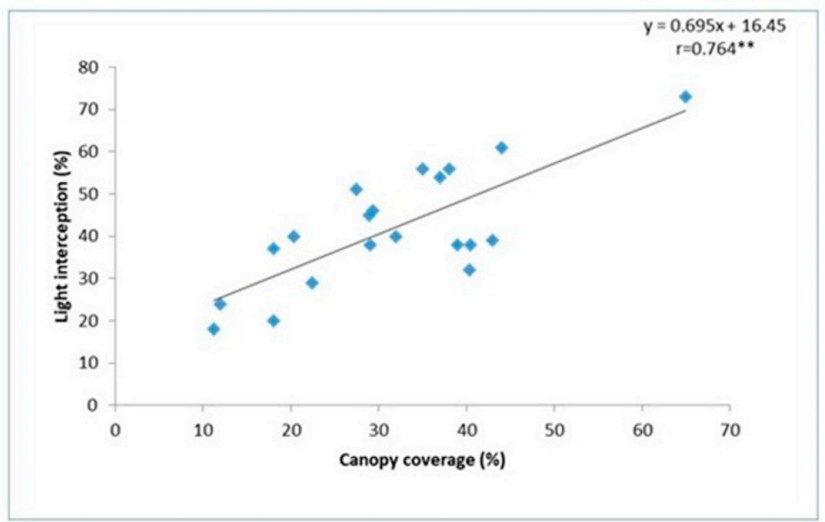

**Figure 3.** Correlation between canopy coverage (%) image acquired in parallel to the ground and light interception. ** denotes significance at 1%.

### 3.3. Canopy Cover in Sugarcane Crop, and Its Correlation with Morphological Parameters, Biomass, and Cane Yield Traits in Field Conditions

The results for CC%, germination %, leaf area (cm$^2$), leaf length (cm), leaf width (cm), leaf number, plant height, and shoot thickness are shown along with LSD at 5% in Table 2.

**Table 2.** Variation in canopy coverage (CC), germination % (G), and leaf and shoot morphology in sugarcane clones under field condition.

| Genotypes | CC% | G% | Leaf Area (cm$^2$) | LL (cm) | L.No | LW (cm) | PH (cm) | SD (mm) |
|---|---|---|---|---|---|---|---|---|
| BO 91 | 18.86 DEF | 50.50 A | 584.12 G | 76.08 EF | 6.00 | 1.70 F | 20.50 EFG | 10.33 |
| Co 0212 | 31.71 ABC | 44.95 ABCDE | 878.93 BCDEFG | 93.25 ABCD | 5.83 | 2.15 E | 23.00 BCDEF | 11.00 |
| Co 0238 | 27.80 BCDE | 41.48 BCDE | 1055.78 ABCD | 92.00 ABCD | 5.58 | 2.59 CDEB | 21.33 DEFG | 12.33 |
| Co 10026 | 27.95 BCDE | 45.00 ABCDE | 1088.68 ABCD | 93.91 ABC | 5.50 | 2.80 AB | 26.08 AB | 12.33 |
| Co 1148 | 18.56 DEF | 41.71 ABCDE | 818.78 CDEFG | 77.83 DEF | 6.00 | 2.30 CDE | 19.08 FG | 11.00 |
| Co 13006 | 15.96 F | 37.77 DE | 772.62 CDEFG | 83.16 BCDEF | 5.33 | 2.19 E | 21.41 DEFG | 12.00 |
| Co 2001-13 | 24.70 BCDEF | 45.32 ABCDE | 1031.05 ABCD | 88.41 ABCDE | 6.08 | 2.54 CDEB | 20.91 DEFG | 12.50 |
| Co 62175 | 26.43 BCDEF | 42.82 ABCDE | 1000.48 ABCDE | 89.33 ABCDE | 5.91 | 2.42 CDEB | 25.00 ABCD | 12.50 |
| Co 740 | 22.75 BCDEF | 47.03 ABC | 865.65 BCDEFG | 79.33 CDEF | 6.08 | 2.24 DE | 21.25 DEFG | 12.66 |
| Co 8021 | 25.83 BCDEF | 46.75 ABC | 966.83 ABCDEF | 89.25 ABCDE | 5.75 | 2.46 CDEB | 24.91 ABCD | 12.16 |
| Co 85019 | 26.03 BCDEF | 44.30 ABCDE | 1057.50 ABCD | 87.08 ABCDE | 5.83 | 2.72 BCD | 23.08 BCDEF | 13.50 |
| Co 86010 | 28.71 BCD | 41.43 BCDE | 946.71 ABCDEFG | 80.08 BCDEF | 5.91 | 2.56 CDEB | 23.33 BCDE | 12.16 |
| Co 86032 | 16.85 EF | 46.52 ABCD | 646.62 EFG | 71.50 F | 5.16 | 2.17 E | 20.50 EFG | 11.66 |
| Co 86249 | 32.80 AB | 50.50 A | 1124.91 ABC | 92.66 ABCD | 5.83 | 2.74 ABC | 24.16 ABCDE | 12.00 |
| Co 94008 | 31.25 ABC | 49.30 AB | 1266.71 A | 92.75 ABCD | 5.75 | 3.16 A | 23.00 BCDEF | 11.66 |
| Co 95020 | 39.50 A | 45.87 ABCDE | 1215.46 AB | 100.25 A | 5.83 | 2.74 ABC | 27.33 A | 12.66 |
| Co 99004 | 26.18 BCDEF | 45.83 ABCDE | 1144.82 ABC | 91.41 ABCDE | 5.75 | 2.70 BCD | 25.66 ABC | 11.66 |
| CoLk 8102 | 22.16 BCDEF | 38.19 CDE | 627.051 FG | 84.33 BCDEF | 5.75 | 1.71 F | 18.83 G | 11.00 |
| CoM 0265 | 30.88 ABC | 37.50 E | 912.43 ABCDEFG | 95.41 AB | 5.41 | 2.30 CDE | 18.50 G | 12.00 |
| CoV 92102 | 20.45 CDEF | 41.25 BCDE | 720.83 DEFG | 80.58 BCDEF | 4.66 | 2.43 CDEB | 21.66 CDEFG | 11.00 |
| Mean | 25.77 | 44.21 | 936.3 | 86.9 | 5.7 | 2.4 | 22.5 | 11.9 |
| LSD@5% | 9.4 | 7.3 | 310.4 | 12.8 | NS | 0.4 | 3.4 | NS |

CC%: Canopy coverage %, G%: Germination %, LL: Leaf length, L.No: Leaf number, LW: leaf width, PH: Plant height, SD: Shoot thickness. NS: Non-significant. n = 3 Values carrying the same letters as superscripts in each column are not significantly different from each other treatment.

### 3.3.1. Canopy Coverage

The mean canopy coverage (CC%) of the sugarcane crop was 25.7%, and the minimum and maximum CC% were 15.9 and 32.8, respectively (Table 2). Among the studied clones, Co 0212, Co 0238, Co 10026, Co 62175, Co 85019, Co 86010, Co 86249, Co 94008, Co 95020, Co 99004, and CoM 0265 were recorded with better canopy coverage of more than 25%, while Co 13006, BO 91, and Co 1148 indicated a poor CC% of less than 17%.

### 3.3.2. Plant Height

The mean plant height of the sugarcane crop was 22.5 cm, and the minimum and maximum plant height were 18.5 and 27.33, respectively (Table 2). Among the studied clones, Co 0212, Co 0238, Co 10026, Co 1148, Co 13006, Co 2001-13, Co 62175, Co 740, Co 8021, Co 85019, Co 86010, Co 86032, Co 86249, Co 94008, Co 95020, and Co 99004 were recorded with better plant height of more than the mean plant height (22.5 cm). The clones, viz., Co 95020, Co 10026, Co 62175, and Co 99004, were observed with significantly better plant height compared to other studied clones.

### 3.3.3. Germination Percentage

The mean data of germination % were 41.25, and the clones, viz., Co 2001-13, Co 86249, Co 94008, Co 10026, CoV 92102, and BO 91, recorded significantly better germination %, while the clones Co 13006, CoLk 8102, and CoM 0265 showed relatively less germination % (<40%) (Table 2).

### 3.3.4. Leaf Area, Leaf Number, Leaf Length, and Leaf Width

The mean leaf length (3rd top visible dewlap) of the sugarcane clones was 80.58 cm, and the clones, viz., Co 95020, Co 0212, Co 0238, Co 10026, Co 94008, Co 86249, Co 85019, Co 8021, and Co 62175, were recorded with significantly better leaf length (>85 cm) than other clones, while Co 86032, BO 91, Co 740, and Co 1148 observed with poor leaf length (Table 2). The mean leaf no. per shoot was 5.7, and non-significant differences were observed among the studied clones. The mean leaf width (3rd top visible dewlap) of the sugarcane clones was 2.4 cm, and the clones, viz., Co 94008, Co 95020, Co 86249, Co 86010, Co 85019, Co 10026, Co 99004, Co 2001-13, and Co 0238, exhibited better leaf width (>2.4 cm), while BO 91 and CoLk 8102 recorded poor leaf width compared to other clones.

### 3.3.5. Shoot Thickness

The mean shoot diameter of the sugarcane clones was 11.9 mm, and the clones, viz., Co 85019, Co 740, Co 10026, Co 0238, Co 62175, and Co 95020, were observed with significantly better shoot diameter, while BO 91, Co 0212, Co 1148, and CoLk 8102 recorded less shoot diameter.

### 3.3.6. Scatterplot Matrix

The scatterplot matrix showing the correlation and frequency counts among the studied parameter, i.e., canopy coverage % (CC), cane yield (CY), dry biomass at formative phase (DWFP), shoot diameter (SD), leaf area (3rd top visible dewlap), leaf area ($cm^2$), leaf width (cm), leaf length (cm), leaf number (L.No), and plant height (PH), is shown in Figure 4. The canopy coverage % data acquired through the image have shown a significantly better correlation with plant height (0.581 **), leaf length (0.853 **), leaf width (0.587 **), and leaf area (0.770 **).

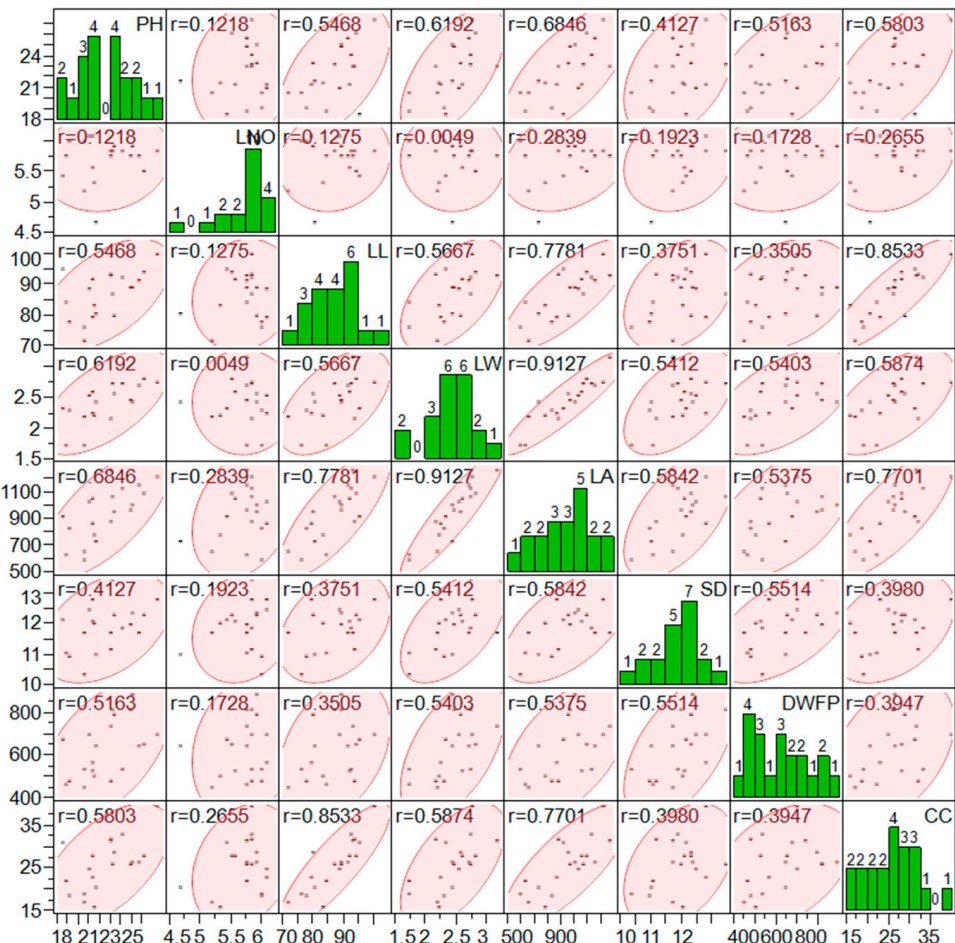

**Figure 4.** Scatterplot matrix showing the correlation, frequency counts among the studied parameter, i.e., canopy coverage % (CC), dry biomass at formative phase (DWFP), shoot diameter (SD), leaf area (3rd top visible dewlap) (LAI), leaf area (cm$^2$) (LA2), leaf width (cm) (LW), leaf length (cm) (LL), leaf number (L.No), and plant height (PH).

### 3.3.7. Two-Way Cluster Analysis

Two-way cluster analysis showing the grouping of sugarcane clones with the studied parameter is shown in Figure 5. The results revealed three distinct clusters: **Cluster I**: BO 91, CoLk 8102, Co 1148, Co 13006, Co 86032, and CoV 92102; **Cluster II**: Co 0212, CoM 0265, Co 0238, Co 86249, Co 10026, Co 99004, Co 94008, and Co 95020; and **Cluster III**: Co 2001-13, Co 85019, Co 62175, Co 86010, Co 8021, and Co 740. Among the three clusters, Cluster I was recorded as relatively lesser in plant height (20.33 cm), leaf number (5.49), leaf length (78.9 cm), leaf width (2.1 cm), total leaf area (695 cm$^2$), shoot diameter (11.2 mm), dry biomass (510 g dry-weight m$^{-2}$), and canopy coverage (18%), while Cluster II was observed with better plant height (23.64 cm), leaf length (94 cm), leaf width (2.65 cm), total leaf area (1085 cm$^2$), and canopy coverage (31%). Cluster III was recorded with better leaf number, dry biomass, and shoot thickness among the studied parameters.

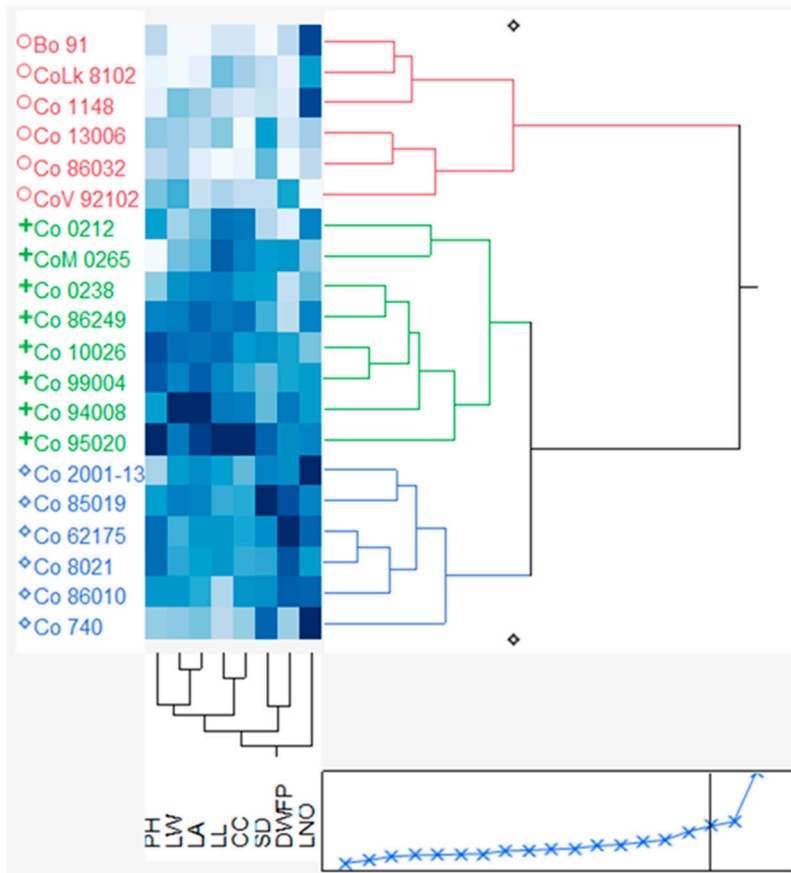

**Figure 5.** Two-way cluster analysis displaying the ward method grouping of sugarcane clones based on the studied parameter.

### 3.4. Canopy Cover in Sugarcane Crop (Improved Breeding Population, Interspecific Hybrid, and Basic Germplasm Clone) and Its Correlation with Physiological, Morphological, and Cane Yield Traits

Canopy coverage:

The mean canopy coverage (CC%) of the sugarcane crop was 32.5%, and the minimum and maximum CC% were 17.2 and 49.0, respectively (Table 2). Among the studied clones, the 004-73, 04-423, 14-161, GUK 06-402, and 01-803 were recorded with better canopy coverage of more than 40% (Figure 2b), while 04-595, 97 GUK 111, and 98 GUK 116 indicated a poor CC% of less than 20% (Figure 6).

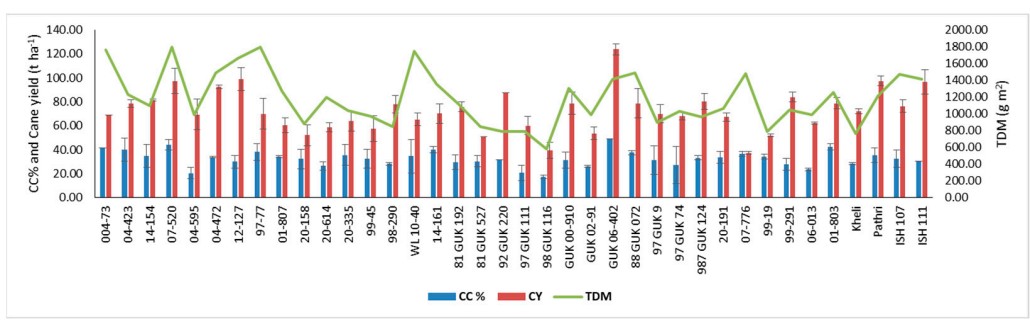

**Figure 6.** The mean cane canopy (CC%) coverage, cane yield (t/ha), and total dry matter (TDM) of various pre-breeding, germplasm, and interspecific hybrid sugarcane clones.

Cane yield:

The mean, minimum, and maximum cane yield in sugarcane clones (improved breeding population, interspecific hybrid, and basic germplasm) were 72.4, 37.3, and 123.8 (t/ha). Among the studied clones, 07-520, 12-127, GUK 06-402, 987GUK 124, 99-291, Pathri, and ISH 111 recorded better cane yield compared to other clones (Figure 6).

Distribution of the dry matter partitioning and physiological and morphological traits:

The distribution of the dry matter partitioning (LWT: leaf weight, SHWT: sheath weight, STWT: stem weight, and TDM: total dry matter (g .dwt.m$^{-2}$) and S.Hgt (cm) in the studied sugarcane clones are shown in Figure 7a. The mean LWT, S.Hgt, SHWT, STWT, and TDM were 349, 191, 251,579, and 1179 g .dwt.m$^{-2}$. The distribution of the SUC%: Juice sucrose, CC: Canopy cover, CCSY: commercial cane sucrose, CT: Canopy temperature (°C) (Figure 7b), CY: cane yield, GC: germination count, LAI: leaf area index, NOC: number of canes, NOL: Number of leaves, SHC: shoot count, and SPAD: Soil Plant Analysis Development ratios are shown (Figure 7b). The mean SUC%, CC, CCSY, CT, CY, GC, LAI, NOC, NOL, SHC, and SPAD were 16, 32, 8.2, 33, 72.4, 12, 2.09, 8.2,77, 61, 23, and 26, respectively. The distribution of the chlorophyll fluorescence ($F_v/F_m$) and total chlorophyll (mg.cm$^{-2}$) are shown in Figure 7c. The mean chlorophyll fluorescence (CFL) and total chlorophyll (TC) were 0.609, and 0.0204, respectively.

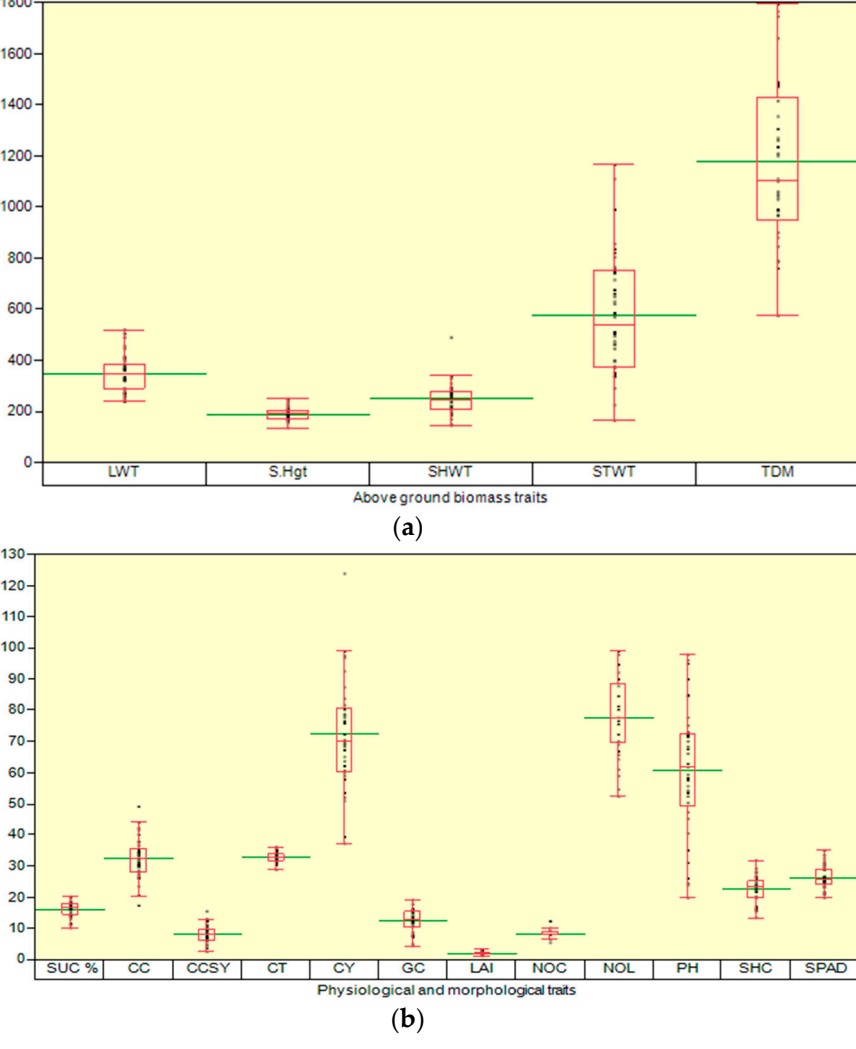

**Figure 7.** *Cont.*

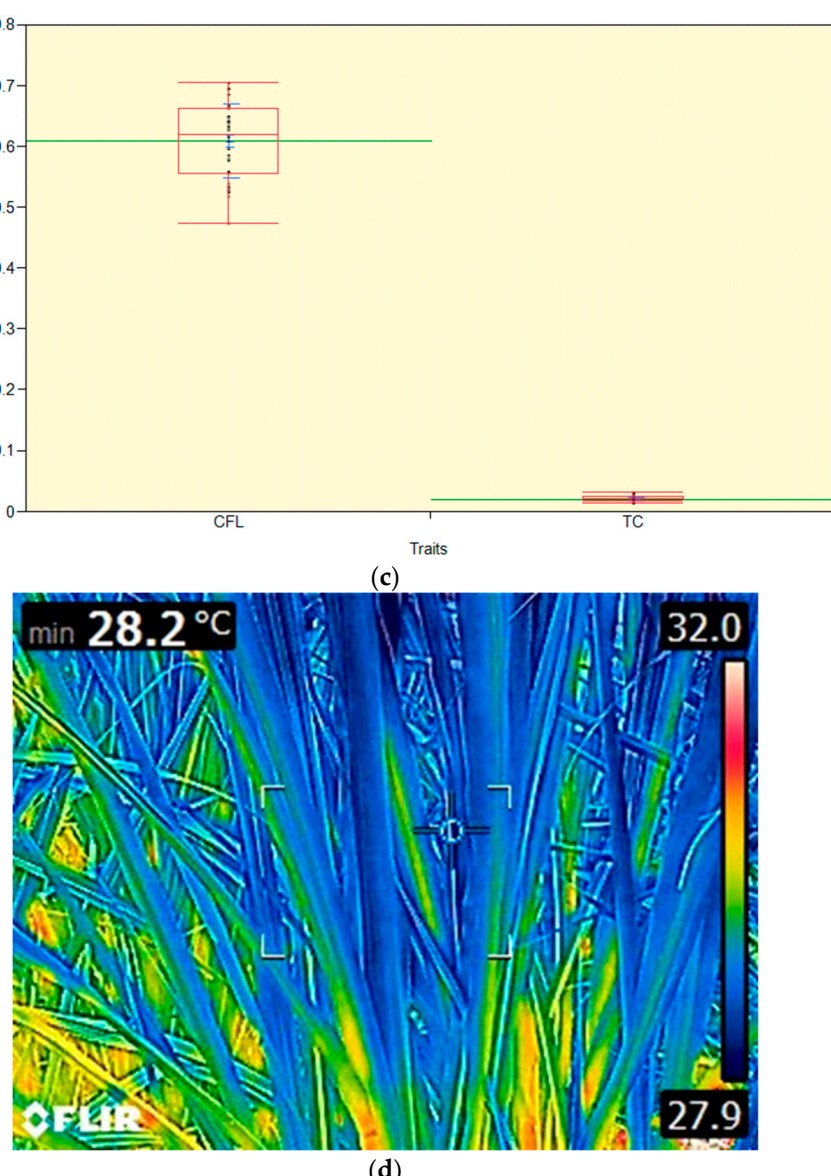

(**c**)

(**d**)

**Figure 7.** (**a**) Box plot displaying (upper quartile, lower quartile, median, upper extreme, lower extreme, whisker, outlier, and mean (horizontal green line) and the distribution of the dry matter partitioning LWT: leaf weight, SHWT: stem weight: STWT, and TDM: total dry matter (g .dwt.m$^{-2}$) ratios) and S.Hgt: shoot height (cm). (**b**) Box plot displaying the distribution of the SUC%: Juice sucrose, CC: Canopy cover (%), CCSY: commercial cane sucrose (ton/ha), CT: Canopy temperature (°C) CY: cane yield (ton/ha), GC: germination count/row, LAI: leaf area index, NOC: number of canes/row, NOL: Number of leaves, SHC: shoot count/row, and SPAD: Soil Plant Analysis Development ratios. (**c**) Box plot displaying the distribution of the physiological (CFL: Chlorophyll fluorescence and TC: total chlorophyll content mg·cm$^{-2}$ ratios). (**d**) Thermal image displaying the canopy temperature (°C) sugarcane crop.

*3.5. Correlation between Physiological and Morphological with Canopy Coverage*

The correlation between physiological and morphological traits and cane yield is shown in Figure 8. Cane yield, SUC%, CCSY, and TDM had significant correlations with canopy coverage (r = 0.46 **, 0.42 **, 0.51 **, 0.62 **, respectively). The germination count, shoot count, initial plant height, and final plant height also had significant correlation with CC (r = 0.46 **, 0.56 **, 0.60 **, and 0.35 *, respectively), while chlorophyll fluorescence ($F_v/F_m$), canopy temperature (CT), SPAD, and total chlorophyll (TC) showed a non-significant association with CC (r = −0.19, 0.00, 0.00, and −0.01, respectively). The

leaf weight and stem weight also revealed a positive correlation (0.36 * and 0.65 **) with CC. Also, the correlation between physiological traits, viz., chlorophyll fluorescence, SPAD, total chlorophyll (TC), and canopy temperature (CT), with cane yield (CY) (r = −0.10 ns, −0.23 ns, −0.23 ns, 0.01 ns, respectively). The leaf area index (LAI), plant height (PH), and total dry matter (TDM) had significant correlations with CC (r = 0.44 **, 0.60 **, and 0.62 **, respectively).

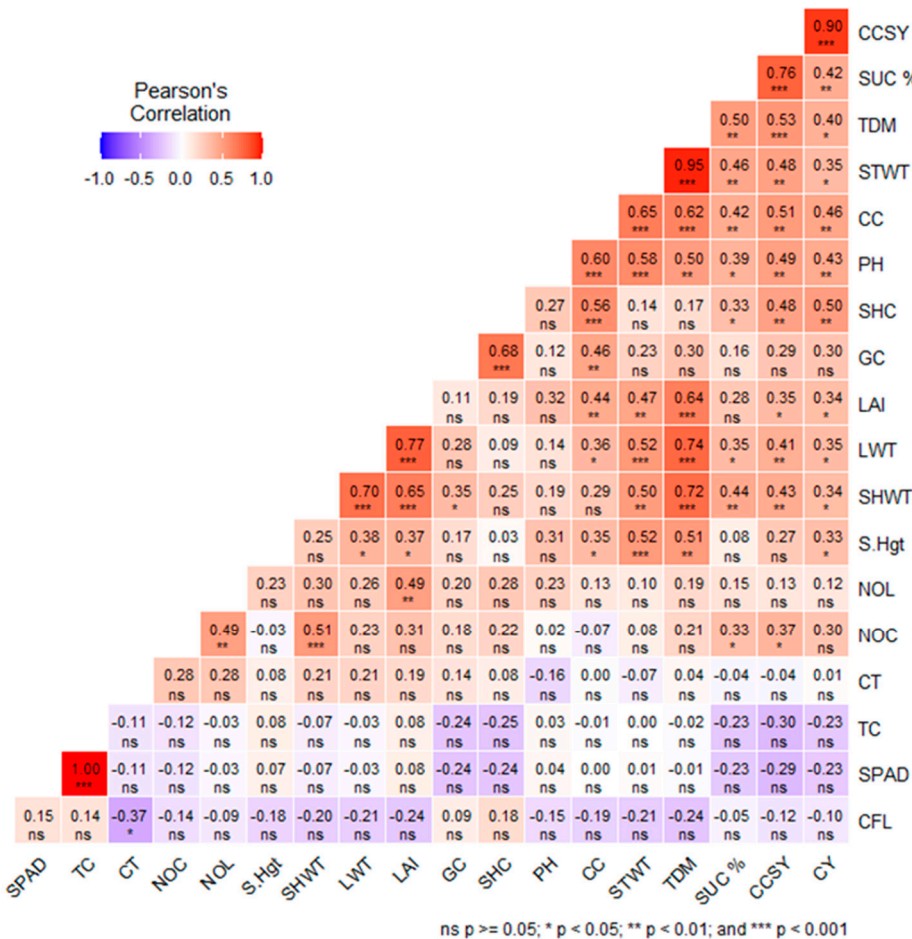

**Figure 8.** Correlation between various physiological and morphological traits with yield, viz., SPAD, TC:total chlorophyll (mg/cm$^2$), CT: Canopy temperature ($^\circ$C), NOC: number of canes/row, NOL: number of leaves, S.Hgt: shoot height (cm) at final stage, SHWT: Sheath weight (g.m$^{-2}$), LWT: leaf dry weight (g.m$^{-2}$), LAI: Leaf area index, GC: germination count, SHC: shoot count-early stage, PH: plant height at early stage (cm), CC: canopy coverage (%), STWT: stalk weight (g.m$^{-2}$), TDM: total above-ground dry matter (g.m$^{-2}$), SUC%: Juice sucrose%, CCSY: Commercial cane sugar (t ha$^{-1}$), and CY: Cane yield (t ha$^{-1}$), *** denotes $p < 0.001$, ** denotes $p < 0.01$; * denotes $p < 0.05$; and ns denotes non-significant $p \geq 0.05$. The intensity of the colour indicates the strength of the correlation.

Based on the canopy coverage data, the simple regression analysis has revealed the prediction of cane yield and total dry matter (TDM) as per Equations (5) and (6) mentioned below:

$$\text{Cane yield} = (33.00 + 1.213\,\text{CC\%}) \tag{5}$$

$$\text{Total dry matter} = (198.18 + 30.204\,\text{CC\%}) \tag{6}$$

These simple Equations (5) and (6) suggest the usefulness of the canopy coverage trait in forecasting cane and total dry matter in sugarcane in a rapid and accurate way.

## 4. Discussion

### 4.1. Position of the Camera and Comparison of Light Interception (LI) with Canopy Cover (CC)

This paper describes the methodology of canopy cover (CC) determination in sugarcane and its association with dry matter and cane yield. A significant correlation (r = 0.870 **) was observed between canopy coverage (%) from images acquired parallel to the ground and perpendicular to the ground (Figure 2). Also, the canopy coverage (%) from images acquired parallel to the ground had a significantly better correlation with the leaf area thus revealing that the parallel position of the camera for capturing the image for CC% is suited for the sugarcane crop. A significant positive correlation coefficient (0.764 **) between light interception (by line quantum sensor) and canopy cover coverage (by *Canopeo*) indicates (Figure 3) the similarity between both data. There is a strong correlation between the canopy coverage (%) image acquired in parallel to the ground and light interception (Figure 3) by a line quantum sensor which we have observed in sugarcane crops also in the present investigation [13]. Similar to our findings, others have also reported that the ground coverage values estimated from digital images taken above the canopy have been correlated to light interception measurements which are limited by the time of measurements and the presence of clouds [25]. The limitation of this light interception method is that the measurements should be taken in unobstructed sunlight and close to solar noon [26]. The canopy cover methodology for estimating light interception in soybeans has been reviewed to have advantages over the above limitations [27]. In this technique, ground area coverage was determined by digital images taken above the canopy. The canopy coverage values were similar throughout the day and were correlated in a one-to-one relationship with light interception measurements made with a line quantum sensor at solar noon. Shepherd et al. (2018) [13] have also reported a linear relationship between canopy cover measured with pictures ($R^2$ = 0.94) and videos ($R^2$ = 0.92) in *Canopeo* and light interception.

### 4.2. Germination

Better germination of sugarcane sets in the field is often reported to be linked with the early vigour. In our study, the mean germination % was 44.21, and the clones, viz., BO 91, Co 10026, Co 740, Co 8021, Co 86032, Co 86249, Co 94008, CoV 92102, Co 95020, and Co 99004, recorded a significantly better germination (>44) percentage. Several reports [28,29] suggest that, due to the genetic nature and environment, there exists high variability in sett germination percent in sugarcane varieties, and these reports corroborate our findings.

### 4.3. Leaf Length, Width, Leaf Number, and Leaf Area

The rate of leaf appearance is cultivar-dependent and determined mainly by temperature [30], but it can also be altered by water stress that decelerates expansive growth [31]. Our experiment also confirms the previous report [30] having greater variability in leaf number which suggests that the variation is mainly due to clonal dependence at ambient conditions. Differential thermal requirements for nine sugarcane cultivars to produce the first leaves and the association of the rate of leaf appearance which has the potential for increasing yield [30] are determined by the extent of genetic variation apart from environmental influence.

Leaf arrangement was associated with higher sugar/metric ton, and selection by breeders for higher leaf area indices and for optimum leaf arrangement is suggested [32]. A significant positive correlation between leaf area index and ground cover in potatoes (*Solanum tuberosum*) under different management conditions has been reported [33], and this shows that the canopy coverage (%) image acquired non-destructively through *Canopeo* software using simple android mobile will be useful in determining the leaf area of the sugarcane crop at an early stage rapidly compared to the conventional destructive methods which consume a lot of labour and other resources. *Canopeo* is faster at calculating a canopy cover percentage and can be easily done while in the field. It took less than 1 min to take

three pictures or one video per plot, and with the line quantum sensor, data collection time per plot was variable due to cloud cover.

### 4.4. Dry Matter Production or Biomass

Most of the better-performing sugarcane clones (Co 86010, Co 85019, and Co 10026) identified in this study had a drought-tolerant parent [1], and, in addition to that, Co 62175, Co 85019, and Co 10026 were high-biomass clones. The poor performance of the clones, viz., BO 91, Co 1148, and CoLK 8102, might be plausibly due to their best suitability to subtropical Indian areas rather than a tropical condition in India, while the clones Co 10026, Co 86249, Co 99004, Co 94008, and Co 95020 are of high biomass type with better leaf area production resulting in better canopy coverage.

### 4.5. Tiller Number and Plant Height

The variability (high tillering and shy tillering) in sugarcane tillering and its relation to sugarcane productivity [34,35] have been widely discussed [36]. It was reported that the number of tillers and plant height at six months after planting are highly correlated with canopy cover ($rg$ = 0.72) and canopy height ($rg$ = 0.69), respectively [37]. Our results are in line with the previous study of [38] which reported that early biomass had a high genetic correlation with unmanned aerial vehicle (UAV)-derived canopy height (0.810) and canopy cover (0.710). Capturing spectral reflectance by means of UAV at the whole canopy level rather than at the individual leaf level has been an important contributing factor for the high trait-yield correlation compared to individual leaf spot measurements which do not represent whole-canopy dynamics [38].

Canopy cover is a useful trait related to crop growth, water use, and stalk number, and cane yield is considered an important parameter in crop monitoring [37].

### 4.6. Canopy Temperature and Cane Yield

Canopy temperature, a surrogate trait for canopy conductance, has been previously monitored in sugarcane, and it showed a significant genotypic variation and a strong negative genetic correlation with biomass [39,40]. Our study (Figures 7b,d and 8) observed similar findings (r = 0.04 $^{ns}$, r = 0.01 $^{ns}$ between CT and TDM, CY, respectively) and also corroborates the report [41] where canopy temperature has been reported as highly negatively correlated with stalk productivity (r = −0.53 **) under drought stress, while there is a non-significant correlation (r = −0.18 $^{ns}$)). Under ambient conditions, the canopy temperature is generally observed with less variability (poor r value with cane yield) among the sugarcane clones, and the better expression of canopy temperature is observed only under abiotic stress conditions where the deeper roots function in tapping of water at deeper zones and support transpiration with subsequent higher canopy conductance, canopy cooling, and better correlation with crop yield.

### 4.7. Chlorophyll Fluorescence vs. TDM and Cane Yield

Chlorophyll fluorescence is being reported to be one of the best traits for screening the healthy crop under abiotic stress and in the present investigation (Figure 3) where the crop responses under ambient conditions did not translate in the form of TDM (r = −0.24 $^{ns}$) and cane yield (r = −0.10 $^{ns}$). The chlorophyll fluorescence exhibits a non-significant correlation (r = 0.02 $^{ns}$) with cane yield under ambient conditions, while a positive correlation of *Fv/Fm* with stalk productivity (r = 0.56 **) under drought stress [41].

### 4.8. SPAD Index vs. TDM and Cane Yield

The SPAD index is a widely discussed trait for the rapid determination of chlorophyll content, and it is also reported to have a significant correlation with crop yield. Chlorophyll is the basic molecule that helps in the absorption of solar radiation and aids in the synthesis of carbohydrates through photosynthesis and finally crop yield. In our experiment, a non-significant correlation of *r* = −0.01 $^{ns}$ and *r*= −0.23 $^{ns}$ was observed between the TDM,

cane yield, and SPAD (Figure 3). These findings corroborate the findings of conclusions of Silva (2007) where the SPAD index has been reported to have a non-significant correlation with stalk productivity ($r = 0.19^{ns}$) under ambient condition, while there is a significant correlation ($r = 0.36^{**}$). Thus, it reveals that the SPAD index is a useful trait preferably under abiotic conditions, where the stress leads to loss of chlorophyll and declined photosynthesis and reduced synthesis of carbohydrates and finally crop yield.

*4.9. Canopy Coverage vs. TDM and Cane Yield*

Canopy cover is a valuable trait for monitoring crop productivity [13], and canopy photosynthesis is greatest when the crop reaches its maximum canopy cover to intercept virtually most of the incident light and absorbs the required photosynthetic radiation for photo-biochemical processes and yield formation [14,15]. From the present study, it is clear that the GUK clones had significantly better CC and cane yield compared to other clones. The GUK clones have the parental genes of *Erianthus sps* which is fast growing, with more leaf area, CC, biomass, and cane yield. It has been reported that *Erianthus sps* exhibits vigorous growth, high biomass production, and high tillering ability and is suitable for abiotic stress conditions [42]. Our experiment results (Figure 8) also confirm the previous reports by displaying significant correlations of $r = 0.46^{**}$, $0.42^{**}$, $0.51^{**}$, and $0.62^{**}$, respectively, of CY, SUC%, CCSY, and TDM with canopy coverage. The germination count, shoot count, initial plant height, and final plant height also had a significant correlation with CC ($r = 0.46^{**}$, $0.56^{**}$, $0.60^{**}$, and $0.35^{*}$, respectively). The leaf weight and stem weight also revealed a positive correlation ($0.36^{*}$ and $0.65^{**}$) with CC (Figure 8). From the overall discussion, it has been found that the plant height, total dry matter (TDM), and leaf area index (LAI) had significant correlation with the cane yield, and the canopy cover data from digital images act as a surrogate for these traits, and further it has been observed that CC had better correlation (Figure 8) with cane yield compared to the other physiological traits, viz., SPAD, total chlorophyll (TC), and canopy temperature (CT).

*4.10. Summary of Key Findings, Advantages, and Limitations*

4.10.1. Key Findings

In the present investigation, the canopy covering digital images of sugarcane crop by using *Canopeo* software was evaluated for its correlation with the physiological and morphological parameters and cane yield production. The results show that among the studied parameters, canopy coverage had a significantly better correlation with the plant height ($0.581^{**}$), leaf length ($0.853^{**}$), leaf width ($0.587^{**}$), and leaf area ($0.770^{**}$) in commercial-type sugarcane clones (Figure 4).

Canopy cover data of sugarcane clones (improved breeding population, interspecific hybrid, and basic germplasm) also revealed a significant correlation of $r = 0.46^{**}$, $0.42^{**}$, $0.51^{**}$, and $0.62^{**}$, respectively, of cane yield (CY), juice sucrose (SUC%), commercial cane sugar yield (CCSY), and total dry matter (TDM) with canopy coverage (CC). The germination count, shoot count, initial plant height, and final plant height also had a significant correlation with CC ($r = 0.46^{**}$, $0.56^{**}$, $0.60^{**}$, and $0.35^{*}$), respectively, while the chlorophyll fluorescence, canopy temperature, and SPAD index revealed a poor correlation with TDM and cane yield. The leaf weight and stem weight also revealed a positive correlation ($0.36^{*}$ and $0.65^{**}$) with CC (Figure 8). From the overall discussion, it has been found that the plant height, total dry matter (TDM), and leaf area index (LAI) had a significant correlation with the cane yield.

4.10.2. Advantages

The traditional light interception method for determining canopy coverage using a line quantum sensor also had a significant positive correlation ($r = 0.764^{**}$) with canopy coverage captured through *Canopeo*; thus, our results signify the importance of canopy coverage determination by *Canopeo* in a rapid, non-destructive way and low-cost way.

### 4.10.3. Limitations

The presence of weeds in the crop field background poses difficulty to classifying or differentiating the crop and weed, and for measuring the canopy coverage, the crop should be in a completely weed-free as well as also detrashed field (removal of senescence leaf) which is more suitable to avoid overestimation of the canopy coverage. If the camera lens were nearer to the crop, then the canopy few crop portions may be excluded in the analysis, and on the other hand, extra sugarcane rows would have been included in the image if the camera lens were positioned at a greater height above the top of the canopy. The vegetation taller than about 2.5 m requires the use of aerial images or special equipment [19].

### 4.11. Future Research Direction

The canopy coverage data measurement through the drone/unmanned aerial vehicle-based image and the utilization of pix4d software version 4.8.4 and other software are an emerging trend for the determination of canopy coverage which is valuable for yield forecasting in sugarcane and other crops.

## 5. Conclusions

The present investigation revealed that in commercial sugarcane clones the mean data of canopy cover were 25.77%, and the clones, viz., Co 95020, Co 0212, CoM 0265, and Co 86249, showed significantly better canopy cover % (>30%) compared to other clones, while the clones Co 13006, BO 91, and Co 1148 were observed with poor canopy coverage (<20%). Also, among the observed traits, canopy coverage % data acquired through image have shown a significantly better correlation with the plant height (0.581 **), leaf length (0.853 **), leaf width (0.587 **), and leaf area (0.770 **). Further, there is a significant correlation (r = 0.585 **) between the canopy coverage (%) image acquired in parallel to the ground and the light interception through line quantum sensors which consume more labour and costly instruments/sensors. The canopy coverage (%) image acquired non-destructively through using simple Android mobile will be useful in determining the leaf area of the sugarcane crop at an early stage rapidly compared to the conventional destructive methods which consume a lot of labour and other resources. Two-way cluster analysis revealed that Cluster II comprising Co 0212, CoM 0265, Co 0238, Co 86249, Co 10026, Co 99004, Co 94008, Co 95020 Co 0238, Co 86249, Co 10026, Co 99004, Co 94008, and Co 95020 was observed with better plant height (23.64 cm), leaf length (94 cm), leaf width (2.65 cm), total leaf area (1085 cm$^2$), and canopy coverage (31%). In a second field experiment with diverse sugarcane clones (improved breeding population, interspecific hybrid, and basic germplasm), the canopy coverage showed a significantly better correlation with biomass (r = 0.612 **) and cane yield (r = 0.458 **), while the chlorophyll fluorescence, canopy temperature, and SPAD index revealed a poor correlation with TDM and cane yield. Light interception determined using a line quantum sensor had a significant positive correlation with canopy coverage signifying the importance of canopy coverage determination in a non-destructive way.

**Author Contributions:** Project formulation and execution of the experiment: R.A.K., S.V., G.H., K.V., S.A. (Shareef Anusha), S.A. (Srinivasan Alarmelu), K.M. and V.S.; Canopy cover data analysis: R.A.K., P.G. and C.P.; Data analysis: R.A.K. and M.A.; Physiological analysis: R.A.K., K.H., V.V. and R.V.; drafting of the manuscript: R.A.K., V.S., K.V., C.A., A.S.T., R.G., P.D. and M.R.M. All authors have read and agreed to the published version of the manuscript.

**Funding:** This research received no external funding.

**Institutional Review Board Statement:** Not applicable.

**Data Availability Statement:** The data presented in this study are available on request from the corresponding author.

**Acknowledgments:** The authors are thankful to the Director, ICAR-Sugarcane Breeding Institute, Coimbatore, for the constant encouragement and support for carrying out the research work.

**Conflicts of Interest:** The authors declare no conflict of interest.

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
