# Peer review of "Rapid and Non-Destructive Methodology for Measuring Canopy Coverage at an Early Stage and Its Correlation with Physiological and Morphological Traits and Yield in Sugarcane"

_agriculture, doi:10.3390/agriculture13081481_

Round 1
Reviewer 1 Report
Dear authors,
Similar work in sugarcane has already been published by several groups with a few examples here (1. https://www.researchgate.net/publication/299820903_Canopy_temperature_a_predictor_of_sugarcane_yield_for_irrigated_and_rainfed_conditions; 2. https://www.cropj.com/carsaroli_14_3_2020_400_407.pdf and 3. hindawi.com/journals/ija/2016/2561026/#materials-and-methods). Therefore, it lacks novelty to be published in a prestigious journal (Agriculture). Therefore, my decision is to reject the article.
Not applicable
Author Response
Sir, Please see the attachment

Reviewer 2 Report
This paper investigates the importance of rapid and non-destructive screening for elite sugarcane genotypes in accelerating varietal/clonal selection. This paper is already a well-developed work. The background is adequately introduced. The methodology is clearly presented. The result is thoroughly presented and discussed. The English writing is good.
The only issue I suggest the author may improve is the discussion section. The current discussion section mainly analyzes the experiment results. While the current analysis of the experimental results is solid, there is room for summarizing discussion on key findings, advantages, and limitations of the proposed methodology for measuring canopy coverage. Moreover, potential improvements and future research directions would add substantial value, thereby rendering the paper even more insightful and forward-looking.
Author Response
Sir, Please see the attachment

Reviewer 3 Report
Dear Authors,
This study examines the correlation between canopy cover (CC) with physiological, and morphological traits in sugarcane. Canopeo software was employed to estimate the CC.
However, there is a lot of clarity is needed in the methodology.
1. Line 140: Why particularly 60DAP is chosen alone for the CC measurement? explain the scientific basis.
2. what is the distance between the mobile and plant while recording the measurements? explain in detail how the measurement was taken.
3. What is the accuracy of the CC recorded through Canopeo software?
4. Have you done any validation of the CC recorded through Canopeo software with the field-measured canopy cover through another approach? Since your study is correlating CC with physiological, and morphological traits measurement, it is important to assess the accuracy of CC measured by Canopeo software.
5. What is the time of observation? How many sample recordings are collected? The represented values are the average of how many measurements?
Overall, I suggest to rewrite the core methodology section in detail. Also, add the accuracy of the Canopeo CC value to field-measured values in the result section.
Author Response
Sir, Please see the attachment

Round 2
Reviewer 1 Report
Dear Editor,
I am not satisfied with the rebuttal by the authors so please reject the article.
Author Response
Sir, Please see the attachment.

Reviewer 3 Report
-
Author Response
Sir, pl see the attachment
